# Mathematical Model and Synthetic Data Generation for Infra-Red Sensors

**DOI:** 10.3390/s22239458

**Published:** 2022-12-03

**Authors:** Laura Leja, Vitālijs Purlans, Rihards Novickis, Andrejs Cvetkovs, Kaspars Ozols

**Affiliations:** Institute of Electronics and Computer Science, 14 Dzerbenes St., LV-1006 Riga, Latvia

**Keywords:** infrared sensors, synthetic data, calibration, microbolometer, non-uniformity

## Abstract

A key challenge in further improving infrared (IR) sensor capabilities is the development of efficient data pre-processing algorithms. This paper addresses this challenge by providing a mathematical model and synthetic data generation framework for an uncooled IR sensor. The developed model is capable of generating synthetic data for the design of data pre-processing algorithms of uncooled IR sensors. The mathematical model accounts for the physical characteristics of the focal plane array, bolometer readout, optics and the environment. The framework permits the sensor simulation with a range of sensor configurations, pixel defectiveness, non-uniformity and noise parameters.

## 1. Introduction

Nowadays, infrared (IR) imaging systems are of particular interest since they support a wide range of applications in many fields across civilian and military use-cases, i.e., IR imaging enables night vision, significantly improves the safety of autonomous systems and is a good candidate for sensor fusion [1]. The development of IR sensors has a growing tendency—it brings improved comfort and size, reduces costs and optimizes detection capabilities [2,3,4].

One of the key challenges to further improve sensor capabilities is the development of efficient algorithms for IR data pre-processing [4,5]. The main goal of the research presented in this paper is to enhance IR camera calibration and design by emulating image sensor raw data that contains side effects of non-uniformity fixed pattern noises (FPN) and averts different kinds of dead and defective pixels, so that the resulting image after processing is appropriate for further use.

To achieve this goal, we have created a mathematical model of IR image sensor and corresponding software (https://github.com/edi-riga/applause-ir-modelling) (accessed on 30 November 2022) that imitates data of a real sensor with FPN and defective pixel (DP) side effects. The output data of the mathematical model was used to apply and adjust existing FPN and DP correction algorithms and to develop a complete calibration procedure of an IR image sensor, for example, for register transfer level (RTL) implementation and variety of sensor configurations. The mathematical model also provides flexibility to emulate raw data of different image formats: VGA, HD, etc.

From the physical perspective, an IR sensor has both external and internal noise sources. Internal noise sources are mainly the atoms of the IR detector, which can vibrate slightly even at low temperatures, and elementary particles that spread and move in a vacuum at a variable rate. External noise sources are ambient temperature fluctuations, electrical supply and related disturbances. Regardless of the heat generated by the IR camera, which can be determined during manufacturing and measured using an integrated thermal sensor, the properties of the IR detectors may change over time, affecting the image quality due to ambient temperature fluctuations [4,6].

Still, one of the main challenges in IR imaging is to achieving continuous correction of non-uniformity in real-time. The IR detector does not have a smooth temperature-impedance relation, and the IR readout circuit’s amplifier has a non-uniform current–voltage characteristic; further, to compensate for the effect of the sensor’s temperature, the readout circuit usually amplifies the difference between active and hidden bolometer readings, therefore producing horizontal or (less often) vertical lines in the raw images. The main factors causing noise are the variation in the bolometer responsiveness and the non-uniform radiation produced by the detector. These noise effects manifest as undesirable image artifacts that human observers are particularly sensitive to [4]. Furthermore, the computational complexity for uncooled thermal imaging systems is still the main factor that reduces the imaging performance and encumbers further image processing tasks, such as image fusion [4,5].

In order to account for the FPN, it is necessary to carry out a non-uniformity correction (NUC). NUC techniques have several methods, but the main ones are generally classified into two categories: scene-based methods and calibration-based methods [7]. The scene-based methods require that camera moves by a specific field of view (FOV) angle during the correction procedure, and the scene-based NUC methods are too complex to implement in the camera’s on-board processing hardware [8]. Here, we utilize the calibration-based methods due to simplified integration and implementation, also reducing latency, which is crucial for real-time control systems [9,10]. To accurately calculate the gain and offset parameters for each pixel’s detector, black-body calibration must be performed at different sensor temperatures [7,11].

Another consideration for the image quality is the correction of defective pixels. The production of the different microbolometer arrays in the IR cameras results in discrepant characteristic curves and noise for all pixels, so the pixel that is significantly different from the mean is defined as non-functional or defective [12,13]. Defective pixels are easy to spot because they are usually isolated and have radically different readings, degrading the image quality. There are several types of defective pixels, but the most common causes are excessive and low responsivity. Excessively responsive pixels always output a near-full-scale pixel value, while low responsivity pixels consistently return a value close to zero [14].

Regardless of the noise caused by the imaging sensing technology (microbolometer), additional inaccurate pixel values emerge due to defects in other manufacturing processes or due to signal conversion. Simultaneously, the sensor may unevenly overheat due to internal losses further affecting data readout. Therefore, calibration is required to prevent discrepancies between pixel voltage and observed temperature [14,15].

## 2. Related Work

To get rid of the above-described non-uniformity (FPN, DP, strip noise) many techniques have been developed in recent years on the correcting algorithms for infrared images.

One of the alternatives to reducing the cost of an IR camera is to produce an uncooled sensor that consists of a microbolometer focal plane arrays (FPAs) that operate at room temperature. Tempelhahn et al. [16] proposed a solution by developing a compensation method with calibration measurements at ambient temperatures without the thermal sensor stabilization to obtain the uncertainty of the low-temperature measurements. They also compared the results with the shutter-based approach, showing that the residual absolute temperature deviations are significantly lower than for a camera with a shutter, but the disadvantages of this type of approach are the calibration time (≈11 h) and a large number of continuous correction factor pixels (up to 1024 × 768).

Liu et al. [5] also worked on the challenge of shutter-less uncooled IR cameras by combining the features of scene-based and calibration-based methods. The advantage of their method is the stability against changes in FPA temperature, but it requires the change of the target scene. The proposed algorithm competes by requiring less computational resources and memory; nonetheless, its utilization in real-time applications is challenging because stable NUC factor estimation requires multiple image frames.

The specific aspect of readout amplifier-generated vertical or horizontal strips is addressed by Cao et al. [4], who propose an efficient NUC algorithm for removing strip noise without disrupting the fine detail of the image. The initial original images are decomposed using wavelet decomposition into three scale levels: large, medium and small, where each level consists of additional directional components. The small scale vertical component contains a lot of strip noise and better removes noise with a small filter. In contrast, using a large vertical component with a large controlled filter allows to save image details more efficiently. The results show that by using such a multi-level filtering scheme, the proposed method can better eliminate strip noise and protect the image from blurring compared to other algorithms [17,18,19,20].

Budzier and Gerlach [13,21] described a calibration process of uncooled infrared cameras based on microbolometers for accurate temperature measurement and the mathematical and physical aspects of an uncooled IR camera based on the radiometric model to determine the temperature-proportional output signal replacement. First, the bias voltage of the microbolometer is adjusted as required. Then it corrects the first bad pixels that may have erroneous behavior and performs the NUC, while adjusting the temperature dependence. Further, a second defective pixel correction is performed. This type of calibration is a complex process and requires a uniform ambient temperature. If the ambient temperature changes, a shutter is required.

To avoid factors causing focal plane to drift over time, such as a change in ambient temperature and variation in the transistor bias voltages, Zuo et al. [7] proposed a scene-based NUC method. The proposed scene-based algorithm competes by requiring less computing and memory resources; nonetheless, real-time processing is still a challenge because multiple image frames are required to estimate stable NUC factors.

Notably, there are other alternative calibrating techniques, such as calibration on the focal plane array discussed by Pron and Bouache [22]. However, such methods are suitable only for cooled IR camera data, which are easier to correct, and are not applicable for uncooled thermal sensors. This work aims to reduce the cost of IR cameras by assisting new algorithm development and validation for varying FPA characteristics and assisting the development (including) uncooled and shutter-less IR cameras.

This work also assists algorithm design for real-time performance and energy efficiency. Further tailoring of the acquisition and sensor degradation models supports online calibration using a range of input variables (sensor temperature, ambient temperature, degradation indicators and others), i.e., enables robust shutter-less cameras. The proposed modeling approach can be combined with modern AI-based methods [23], e.g., object tracking task for infrared images [24] as well as to improve quality assurance and inspection applications such as infrared thermography [25], drones for the prevention of fire detection [26], material defect detection [27] and even wind turbine erosion detection [28] while incorporating physical models of different materials.

## 3. Data Acquisition Modeling

In general, the theoretical basis for the mathematical calibration utilizes the radiometric model of the uncooled IR camera. Temperature transfers in several ways (conduction, convection and radiation), and importantly every warm body radiates heat. Hence, several factors must be taken into account: radiance, absorption properties, lens effects, power distribution over the sensor surface, non-uniformity, *V*-*R* characterization of the microbolometer arrays, and the readout properties.

### 3.1. Radiance and Absorption

Valuable resource describing the fundamentals of the IR model, detector operation and testing is provided by Vincent et al. [6] and by Kruse and Skatrud [29]. Figure 1 illustrates a system where the IR sensor with a lens receives heat from an object characterized by radiant flux—Φ. The body radiates heat, which is absorbed by the pixels of the microbolometer FPA. It is converted into a raw signal voltage that is embodied in the captured image, and its pixel values depend on the radiated flux, apart from the internal camera heat. For the mathematical model, we assume this object to be a black-body (BB),I confirm the format of all text - normal the radiance of which is predictable and corresponds to the Planck’s function curve. Planck’s function is [30]:(1)$$B\left(\lambda ,T\right)=\frac{2hc^{2}}{\lambda^{5}}\frac{1}{e^{hc/\lambda KT}-1},$$
where *h* denotes Planck’s constant [J · s], *c* is the speed of light in a vacuum [m/s], λ is the wavelength [m], *K* is Boltzmann’s constant [J · K^−1^], and *T* is the temperature of the source [K].

The radiant flux emitted by the unit surface area $dA_{1}$ is characterized by the object radiant exitance: $M=\Phi_{1}/dA_{1}$ [W/m^2^], while the IR sensor receives irradiance on the unit surface area $dA_{2}$: $E=\Phi_{2}/dA_{2}$ [W/m^2^], where $\Phi_{1}$ and $\Phi_{2}$ denotes the radiated fluxes emitted by the object and received by the sensor. BB has a characteristic radiance *L* (shown in Figure 2, Left ) that indicates how much of the surface radiated power is received by an optical system observing from a certain viewing angle. In Figure 2, Right, the relationship is illustrated when the power is integrated per unit area in a temperature range (from 300 to 400 K) and an almost linear relationship is represented. The BB radiance *L* at a known temperature and for limited band, is calculated by integrating Planck’s Law (from Equation (Equation 1)):(2)$$L=2hc^{2}\varepsilon_{t}\int_{\lambda_{1}}^{\lambda_{2}}\frac{d\lambda }{\lambda^{5}\left(e^{\frac{hc}{KT\lambda }}-1\right)},$$
where $\varepsilon_{t}$ is transmitted BB emissivity and for any particular system (Figure 1), the limits $\lambda_{1}$ and $\lambda_{2}$ are determined by the spectral range of the sensor. Further, in order to illustrate the developed software, we will utilize a wavelength λ range from 8 to 15 μm. We can transform the expression (Equation 2) into numerical series [31] integrating from zero to a specifically chosen wavelength λ:(3)$$L=\int_{0}^{\lambda }Ld\lambda =C_{1}C_{2}^{-4}\frac{T^{4}}{\pi }\sum_{n=1}^{\infty }\left[\frac{e^{-nx}}{n}\left(x^{3}+\frac{3x^{2}}{n}+\frac{6x}{n^{2}}+\frac{6}{n^{3}}\right)\right],$$
where x=hc/(λKT), constants of integrals: $C_{1}=2\pi hc^{2}$ and $C_{2}=hc/K$. The expression for in-band wavelength range integral of blackbody radiance is calculated as follows:(4)$$\begin{array}{c} \begin{array}{cc} \int_{\lambda_{1}}^{\lambda_{2}}Ld\lambda & =\int_{0}^{\lambda_{2}}Ld\lambda -\int_{0}^{\lambda_{1}}Ld\lambda \\ \; & =C_{1}C_{2}^{-4}\frac{T^{4}}{\pi }(\sum_{n=1}^{\infty }\left[\frac{e^{-nx_{2}}}{n}\left(x_{2}^{3}+\frac{3x_{2}^{2}}{n}+\frac{6x_{2}}{n^{2}}+\frac{6}{n^{3}}\right)\right]\\ \; & \;-\sum_{n=1}^{\infty }\left[\frac{e^{-nx_{1}}}{n}\left(x_{1}^{3}+\frac{3x_{1}^{2}}{n}+\frac{6x_{1}}{n^{2}}+\frac{6}{n^{3}}\right)\right]), \end{array} \end{array}$$
where $x_{1}=hc/\left(\lambda_{1}KT\right)$, $x_{2}=hc/\left(\lambda_{2}KT\right)$. We can transform constant before the integral into simpler view: $C_{1}C_{2}^{-4}T^{4}/\pi =2K^{4}T^{4}/\left(h^{3}c^{2}\right)$. The tolerance of the result depends on count of summing row members—*n*.

### 3.2. Lens Effects

Sensor optics are described by the principal planes and focal points. The size of the scene *S* is determined by the field of view (FOV) and the distance *D* of the scene to the principal plane of the optics:(5)$$S=2D\tan\frac{\mathrm{FOV}}{2}.$$

Likewise, in the scene, object size *A* radiating a single pixel is given by its instantaneous field of view (IFOV):(6)$$A=2D\tan\frac{\mathrm{IFOV}}{2},$$
then expressing FOV (seen Figure 3), which depends on the sensor size *l* and focal length *f*, is:(7)$$\mathrm{FOV}=2\arctan\frac{l}{2f}.$$

However, for a single pixel with a pixel size of *a*: IFOV =2arctan(a/(2f)) [32].

Figure 4 illustrates the path of the radiation from an object-side area element $dA_{1}$ through the entrance pupil of the optics to the image-side area element $dA_{2}$. We assume that the sensor “observes” the scene via a round aperture, and the projected solid angle of a round aperture is:(8)$$\Omega =\pi \sin^{2}\phi_{2}.$$

Assuming a lossless environment, the object side radiation flux $\Phi_{1}$:(9)$$\Phi_{1}=dA_{1}\pi L_{1}\omega_{0}\sin^{2}\phi_{1},$$
is identical to the image side radiation flux $\Phi_{2}$, i.e., $\Phi_{1}=\Phi_{2}$. For convenience and unit adjustment, we can assume $\omega_{0}=1\left[sr\right]$. Therefore:(10)$$dA_{1}L_{1}\sin^{2}\phi_{1}=dA_{2}L_{2}\sin^{2}\phi_{2}.$$

Finally, the image-side irradiance *E* is calculated as:(11)$$E=\frac{\Phi_{2}}{dA_{2}}=\pi \tau L_{2}\omega_{0}\sin^{2}\phi_{2},$$
where τ models the transmission losses in the optics.

### 3.3. Power Distribution

The power distribution across the sensor surface is a continuous function, i.e., the cosine 4th power-law [33] applicable to surface area and also for every pixel. The surface of the pixel closer to the center of the sensor receives more power flow than the one further away. In order to elucidate IR power $P_{t}$ distribution on the surface area *A* of the sensor, first, we determine IR power on a single center pixel—$P_{pix}$—on the surface of the sensitive area of this single pixel $A_{sens}$.

We apply the law of cosine 4th (illustrated in Figure 5) power only within the entire sensor area, as we calculate the discrete values of the power flow that reaches every pixel, depending on its distance from the center of the lens and the center of the sensor. For this purpose, calculating $\cos^{4}\beta$ is required for every pixel, where the angle β is formed by the center of the sensor and a point on the optics. We utilize this data by multiplying every value with $P_{pix}$. The result is a power distribution over the sensor area for every pixel. Therefore, the area of a single center pixel is used as a reference.

Assuming that the reference pixel is located in the middle of the sensor, its received IR power can be determined using the equation:(12)$$P_{pix}=\pi LA_{sens}\sin^{2}\phi_{2},$$
where ϕ—angle of FOV, but $A_{sens}$—the sensitive area of this pixel. Using this value we can calculate IR power distribution for every pixel of the sensor, by utilizing the “natural vignetting” (cosine to the 4th law):(13)$$E_{pix}=\frac{d\Phi_{12}}{dA_{2}}=L\int_{A_{1}}\frac{\cos^{4}\beta }{d^{2}}dA_{1}.$$

Figure 6 illustrates how the power reaches a specific pixel, where point *O* is the exit pupil’s center of the lens. The small horizontal squares represent the pixels of the sensor. The red square represents the single-pixel area at the center of the sensor. The distance *l* between point *O* and the center of the sensor is also the focal length. Further, we can compute the distances between the point *O* and every pixel center, e.g., point *P*. The angle β is formed by the pixel, point *O* and sensor’s center.

In our model we assume that IR power incident on any pixel-sensitive surface is reduced by $\cos^{4}\beta$ factor in comparison with the IR power incident on the red square area in the center of the sensor; hence, Equation (Equation 12). Figure 7 illustrates the sensor pixel power distribution corresponding to blackbody temperatures from 300 K to 500 K with 25 K increments, we use previously calculated value of in-band wavelength range integral of blackbody radiance at given temperature and the optical parameter-focal length is taken 0.0212 mm.

### 3.4. Mircobolometer Characterization

A microbolometer is characterized by temperature *T* dependent electrical resistance and essentially is a thermal sensor that receives radiation from the optical system with its front surface and depends on changes in electrical resistance *R* to measure the heating effect due to IR radiation. Microbolometers are separated from the thermal reservoir silicon substrate by electrically conductive microbolometer support beams. The microbolometers are installed in a vacuum housing to reduce their sensitivity to temperature changes and improve stability, and accordingly, this reduces the sensor’s temperature deviations due to heating and changes in ambient temperature. Usually, the resistance is measured with a constant current, which is supplied only during measurement. Notably, the current heats up the bolometer and further affects the electrical resistance. This relationship can be described by the microbolometer heat balance equation:(14)$$C\frac{dT}{dt}=I^{2}R\left(T\right)+\underset{Q}{\underset{⎵}{\varepsilon P_{t}+\varepsilon P_{s}-\left(2A\right)\varepsilon \sigma T^{4}}}-g\left(T-T_{s}\right),$$
where *C* is heat capacitance [J/K], *T* is the microbolometer’s temperature dependent on the exchange of energy [K], $T_{s}$ is the underlying substrate temperature of the microbolometer [K], *I* is the electric measurement current [A], *R* is the temperature-dependent microbolometer resistance [Ω], *Q* refers to all radiation input power (ε—emissivity, $P_{t}$—*IR* power incident on microbolometer from target, $P_{s}$—*IR* power from surroundings, $2\left(A\right)\varepsilon \sigma T^{4}$—Stefan’s law), and *g* is thermal conductance between the bolometer and its supporting structure [W/K].

The microbolometer heat balance Equation (Equation 14) in thermal equilibrium is:(15)$$Q+I^{2}R\left(T\right)-g\left(T-T_{s}\right)=0,$$

Converting equation and expressing via the temperature difference obtains:(16)$$T-T_{s}=\frac{IV+Q}{g},$$
where V=IR(T)—voltage across the device [V]. *R* temperature dependence is determined by the thermal resistance coefficient: α=dR/RdT. Usually, in the working temperature range, the temperature–resistance relationship is linear, and it can be expressed as ΔR=αTΔT.

Further, the microbolometer’s resistance depends on the temperature as:(17)$$R\left(T\right)=R_{0}\left(T\right)\exp\left(\frac{\Delta E_{a}}{KT}\right),$$
where $\Delta E_{a}$ is activation energy [J] and $R_{0}$ is microbolometer resistance at ambient temperature, i.e., $R_{0}=R_{Ta}/\left(\exp\left(E_{a}/\left(KT_{s}\right)\right)\right)$[Ω]. Respectively, the voltage is given by:(18)$$V=IR_{0}\left(T_{s}\right)\exp\left(\frac{\Delta E_{a}}{KT}\right),$$

For a semiconductor TCR (temperature coefficient of resistance), the steady-state *V*–*I* curve of the microbolometer at temperature *T* is given by coupling Equations (Equation 16) and (Equation 18). Eliminating *T* between these equations gives the *V*–*I* relation:(19)$$V=IR_{0}\left(T_{s}\right)\exp\left(\frac{\Delta E_{a}}{K(T_{s}+\frac{IV+Q}{g})}\right),$$
with the initial conditions:(20)$$\left\{\begin{array}{c} t_{0}=0\\ Q(t_{0})=0\\ I(t_{0})=0 \end{array}\right.\;.$$

### 3.5. Readout

Thus far, the resulting data corresponds to the signal at the input of the *Read-Out Integrated Circuit* (ROIC) that is responsible for multiplexing detectors and producing voltages corresponding to the changes. Figure 8 illustrates the basic functions provided by the ROIC: converting detector signal to voltage, multiplexing, and driving the resulting data streams to external electronics. Modern ROICs may also incorporate on-chip ADCs, limiting susceptibility of the analog signal interference, reducing the overall system size, weight and power [6].

Notably, there is a variety of choices concerning ROIC design: internal or external ADC(s), signal retrieval based on integrator or voltage sampler, signal multiplexing (from unit cells) using voltage or charge, data sampling using snapshot (global) or rolling integration modes, variety of sampling methods, etc. Furthermore, detector may incorporate “hidden” bolometers for sensor’s temperature compensation, which can also be a part of ROIC’s functionality. Thus, modeling is highly dependent on the actual ROIC’s design.

In our model, we assume non-biased readout, i.e., there is no bias current applied to the microbolometer at time $t_{0}$. At the time $t>t_{0}$ the amount of heat generated by applying bias current is:(21)IV(t)t
and the amount of heat generated by IR radiation is:(22)$$Q\left(1-\exp\left(\frac{-\tau }{t}\right)\right)$$

In the current IR simulation model, we use a couple of equations. Equation (Equation 19) represents active pixels (that receive IR power). However, in skimming for pixels, we utilize the same Equation (Equation 19) while neglecting the effect of *Q*. In general, the equation for the microbolometer’s output voltage after measurement time depends on input IR power as follows:(23)$$V\left(t\right)=IR_{0}\left(T_{0}\right)\exp\left(\frac{\Delta Ea}{K[T_{0}+\frac{IV(t)t}{g}+\frac{Q}{g}\left(1-\exp\left(\frac{-\tau }{t}\right)\right)]}\right).$$

To obtain output values of ROIC integrators for each pixel, first, it is necessary to find momentary values and integral equations for active and skimming pixels. For different types of pixels (active, boundary, skimming), we can expand Equation (Equation 19):(24)$$V=I_{B}R_{0}\exp\left(\frac{\Delta Ea}{K(T_{0}+\left(\frac{I_{B}V+Q_{opt}+Qcam}{g}\right)\left(1-\exp\left(\frac{-\tau }{t}\right)\right)}\right).$$

IR power *Q* that is impinged on the pixel consists of IR power sum: radiated via optics—$Q_{opt}$, and IR power radiated by the internal surface of the camera body—$Q_{cam}$, it is: $Q=Q_{opt}+Q_{cam}$. The skimming pixels do not receive any *IR* radiation, so for them—$Q_{opt}=0$. Further, the readout electronics may integrate this voltage using specific electronic circuits, convert the reading to a digital value and perform either analogue or digital temperature compensation.

## 4. Data Synthesis and Results

Variability requirement for the *IR* imaging, sensor composition and readout specifics has caused a design of a model-based simulation framework. The framework is written in Python programming language and is published online under the MIT license. The published repository includes an example configuration using four previously described models: blackbody, optics, bolometers and readout electronics.

Each model is implemented as a derived Python Model class, necessitating it to provide standardized functionality. The model must incorporate a standardized input/output data interface and may be provided with parameterized arguments, where each parameter essentially increases the modelling space by a factor of two. The model may also provide callbacks for ensuring result caching (for faster design space exploration) and output data display. Figure 9 illustrates an example of a high-level simulation. Note that simulation may be expanded with better representations of the real-world complexities, i.e., source may be replaced with an actual (more complex) scene, and an additional model may represent such environmental effects as fog, etc.

The initially suggested model computes emitted power of a black body to the center of camera optics and uses that information to further calculate the power distribution across bolometers, as previously described in Section 3. The bolometer model utilizes Plank’s black body radiation function to integrate the actual power observer by every pixel. The readout model utilizes a theoretical temperature compensation mechanism, where correction is performed within the ROIC electronics. The readout model showcases two approaches where both column- and row-based compensation is used.

Table 1 illustrates the generated output of the simulated model in a variety of configurations. This data can be further utilized to simulate data for a variety of sensor and ambient temperatures, noise sources and sensor’s aging effects, thus making the model useful for algorithm design for uncooled IR sensors.

The “X” axis represents images with a variate degree of randomization for physical parameters, while the images on the “Y” axis correspond to a range of blackbody temperatures. Tolerance is a coefficient by which the nominal value of the correct physical parameter and its obtained part is accepted as one standard deviation for normal distributions. For example, if the nominal value of a physical parameter is $10^{-6}$, and the coefficient is 0.01, then $10^{-8}$ is given as the standard deviation for the parameter. For each temperature and randomization, there are two pictures: one with a single skimming (blind) pixel row, and the other with a skimming column.

One of the next processing steps in thermal imaging cameras usually incorporates some kind of nonuniformity correction (NUC) algorithm. To simplify hardware implementation, linear correction (two-point NUC) with fixed-point binary number representation may be used. When trying to conserve the hardware resources and the memory requirements for correction coefficient storage, the calculation precision choice becomes increasingly important. Table 2 shows simulation results of linear nonuniformity correction with different calibration coefficient fractional part widths. It can be immediately noticed that the microbolometer array’s nonuniformity, although not completely compensated, becomes significantly less apparent even at low coefficient resolutions. Correction of the cosine 4^4th^ power law effect, due to its gradient nature, is more demanding, with insufficient calculation precision causing noticeable circular artefacts.

## 5. Conclusions

Cost reduction of IR imaging solutions is a key challenge, and solving it would increase technologies’ availability and bring benefits to a range of applications, e.g., safety, quality control and system analysis. This work addresses this major challenge by supporting further development of efficient data pre-processing algorithms by reviewing the IR sensor mathematical model, designing a model-based simulation tool and displaying experimental results facilitating the development of algorithms, including shutterless IR cameras.

In general, IR imaging is a complex task, and the development in recent years has brought significantly cheaper and more efficient products. Several studies have proposed different algorithms for uncooled IR cameras’ offline/online calibration and provided a general description of their camera radiometric models. Our proposed data acquisition model approach gives more insight into the relationship between radiance/absorption, IR optics, bolometer design and camera electronics, including readout properties.

The initially developed simulation framework incorporates models for IR blackbody, optics, bolometers and readout electronics. The model permits the generation of simulated data that can facilitate the innovation of calibration algorithms for uncooled IR cameras and account for the runtime degradation of sensors. Notably, the simulation framework is extendable to incorporate additional models (scene, atmospheric effects, degradation, etc.). The framework is published online.

The main advantage of the proposed modeling approach is the ability to parameterize different conditions, thus generating data for representing varying conditions, defining algorithm design challenges and validation. This exploration has been conceptualized by showcasing the parameterization of a fixed-point nonuniformity correction algorithm.

In the future, the modeling and framework can be extended to enable the projection of an actual projected scene, extend the framework to support temporal simulations, develop the model for incorporating other noise sources, add additional readout electronics configurations/models, and simulate such environmental effects as fog. An improved model can be leveraged to further improve physical sensor structure (e.g., cap design, heat dispersion, consequences of the vacuum quality) and readout electronics for increased sensitivity and improved algorithm design (e.g., explore temporal noise filtering while increasing sensor’s readout frequency and employing multi-frame analysis). 

## Figures and Tables

**Figure 1 sensors-22-09458-f001:**
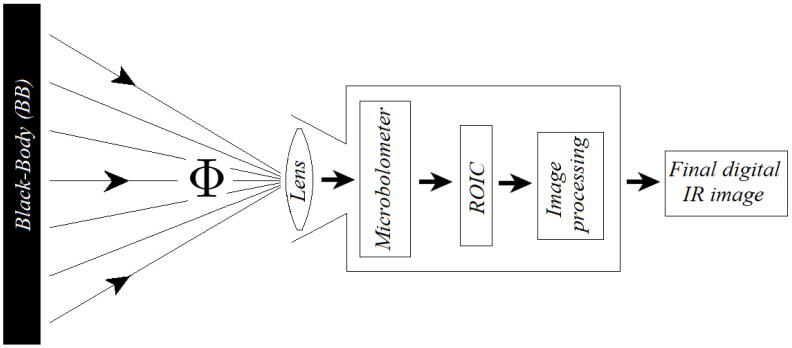
A system where an IR sensor with a lens receives radiant flux Φ from the BB.

**Figure 2 sensors-22-09458-f002:**
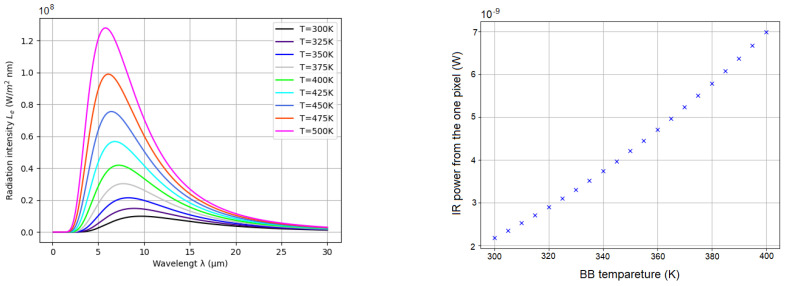
Radiant spectral exitance of BB temperatures from 300 to 500 K. (**Left**): relationship between the BB radiant spectral exitance and the wavelength. (**Right**): relationship between the IR power impugned on one pixel sensitive area, located in the middle of the sensor.

**Figure 3 sensors-22-09458-f003:**
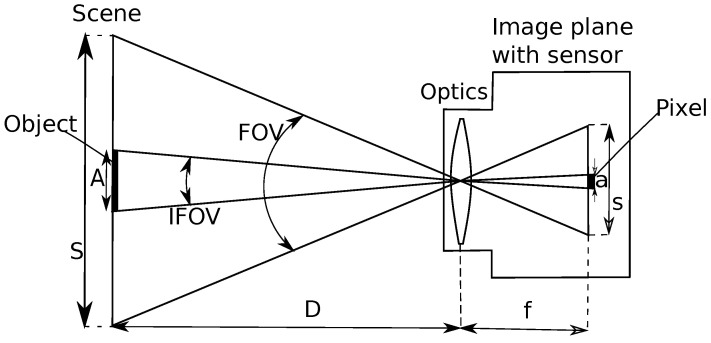
FOV in a sensor array.

**Figure 4 sensors-22-09458-f004:**
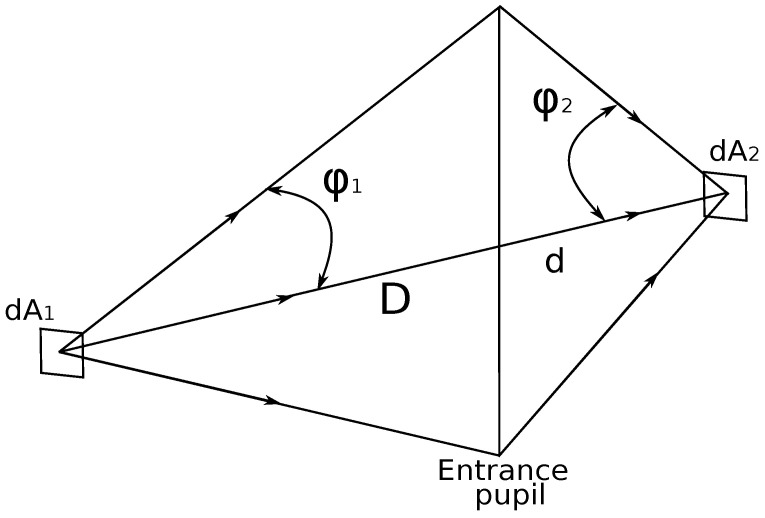
Path of the IR radiation from the object to the image.

**Figure 5 sensors-22-09458-f005:**
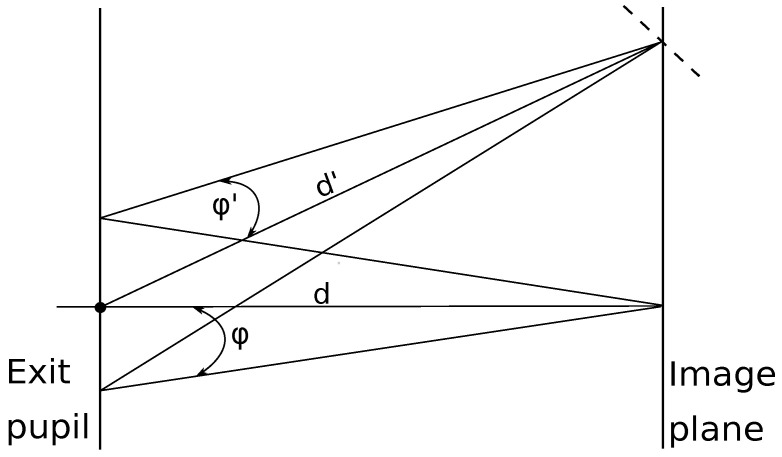
The power flow arriving at the edges of the sensor is smaller by a factor of $\cos^{4}\beta$ compared to the center. The distance $d^{\prime }$ (with the radiation flux $\phi^{\prime }$) needs to travel a longer distance than the distance *d* with the radiation flux ϕ thus, the received power on the center of the sensor is greater than on the edges.

**Figure 6 sensors-22-09458-f006:**
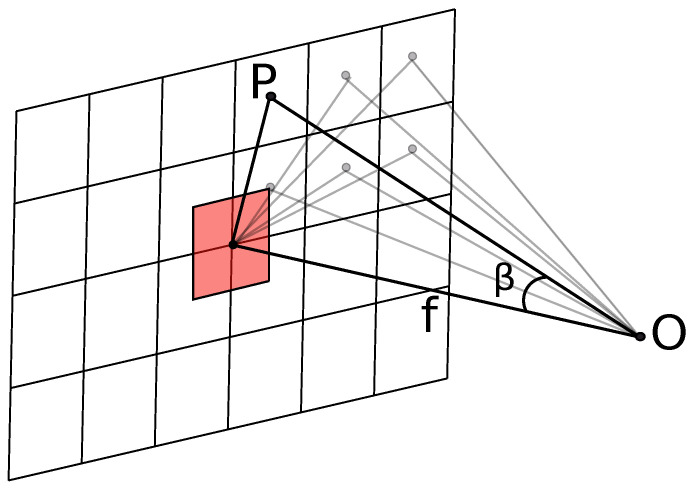
IR sensor power reaching single-pixel.

**Figure 7 sensors-22-09458-f007:**
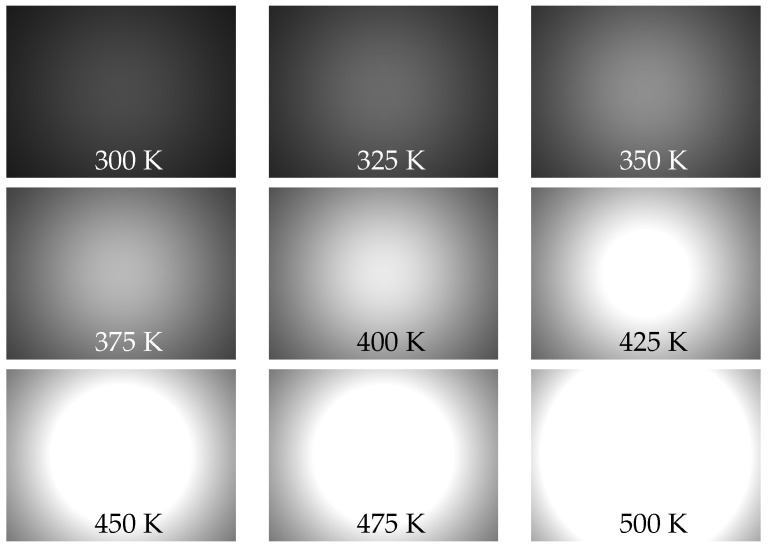
Visualization of IR power distribution over sensor area following $\cos^{4}\beta$ law at different BB temperatures.

**Figure 8 sensors-22-09458-f008:**
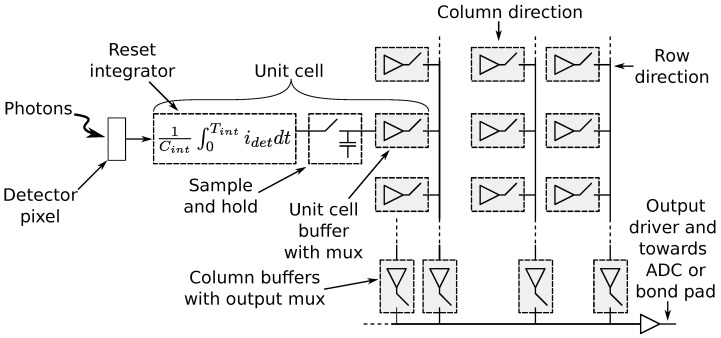
The basic signal path elements within ROIC (integrator, multiplexer, output driver) [6].

**Figure 9 sensors-22-09458-f009:**
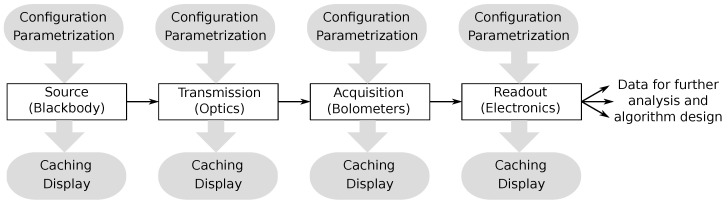
The high-level concept of the Python-based simulation framework.

**Table 1 sensors-22-09458-t001:** Output data of the simulated model at variety of blackbody source temperatures (300–500 K) and bolometer and readout nominal values: microbolometer resistance at ambient temperature ($R_{Ta}=10^{6}\;\left[\Omega \right]$), thermal conductivity ($g=10^{-7}$ [W/K]), thermal capacity ($C=5\cdot 10^{-9}$ [W s/K]) and tolerances (ε). $\sigma_{R_{Ta}}$, $\sigma_{g}$ and $\sigma_{C}$ represent standard deviation of the parameters, where $\sigma_{R_{Ta}}=\varepsilon R_{Ta}$. “rows” and “columns” refer to the used temperature correction method (column or row skimming pixels are used for the compensation).

ε (Tolerance)	10−4	10−5	5·10−6	3·10−6	10−6
Standard Deviations	σRTa= 100, σg= 10−11, σC= 5·10−13	σRTa= 10, σg= 10−12, σC= 5·10−14	σRTa= 5,σg= 5·10−13, σC= 2.5·10−14	σRTa= 3,σg= 3·10−13,σC= 1.5·10−14	σRTa= 1,σg= 10−13, σC= 5·10−15
300 K	rows	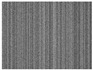		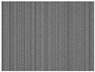	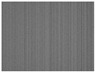	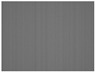
columns	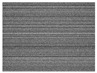	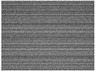	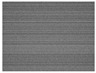	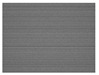	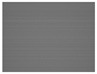
320 K	rows	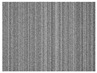		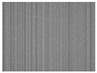	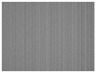	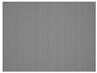
columns	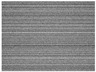	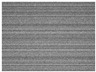	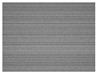	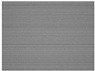	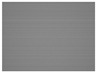
340 K	rows					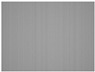
columns	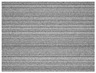	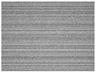	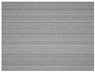	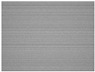	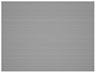
360 K	rows		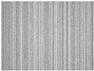	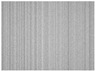		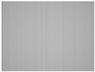
columns	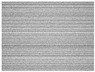	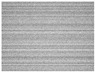	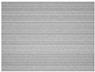	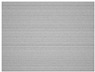	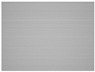
380 K	rows	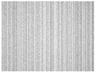	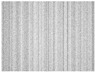		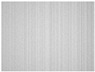	
columns	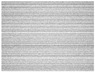	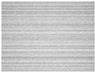	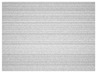	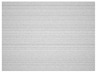	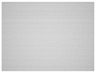
400 K	rows		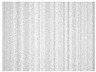		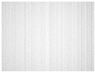	
columns	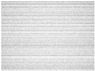	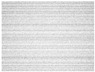	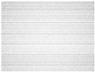	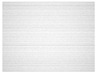	

**Table 2 sensors-22-09458-t002:** Nonuniformity correction with varying fractional part widths of the correction coefficients. Simulation parameters: microbolometer resistance at ambient temperature $R_{Ta}=123\;\left[\Omega \right]$, thermal conductivity $g=10^{-7}$ [W/K], thermal capacity $C=1\cdot 10^{-9}$ [Ws/K] and tolerance $\varepsilon =10^{-5}$.

Fixed-Point Bits in Coefficients	Uncorrected	3 Bits	4 Bits	5 Bits	6 Bits
300 K	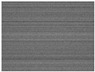	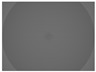	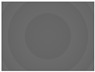	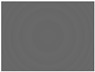	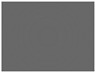
350 K	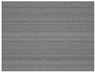	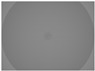	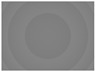	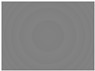	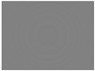
400 K	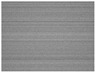	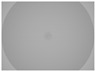	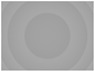	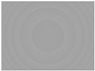	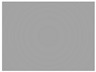

## Data Availability

Developed simulation framework: https://github.com/edi-riga/applause-ir-modelling (accessed on 30 November 2022).

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
