# Peer review of "Mathematical Model and Synthetic Data Generation for Infra-Red Sensors"

_sensors, 2022, doi:10.3390/s22239458_

Round 1

Reviewer 1 Report

This MS addresses the challenge of capability improvement of uncooled IR sensors by providing a mathematical model and synthetic data generation framework.

Although the methods are described adequately with serials of equations and the effectiveness is supported by the experimental results, the background information provided in the MS cannot show the instant application value. So it is recommended to add more recent background reference citations, here are some recommendations:

[1] An Infrared Defect Sizing Method Based on Enhanced Phase Images, Sensors, 2020,

[2] Active Thermography for the Detection of Sub-Surface Defects on a Curved and Coated GFRP-Structure, Applied Science, 2021

[3] Size determination of interior defects by reconstruction of subsurface virtual heat flux for step heating thermography, NDT & E INTERNATIONAL,2023

Besides the above, please check all equations carefully again to ensure they are presented correctly.

Author Response

Thank you for your valuable recommendations. We have checked for the correctness of the mathematical equations (we identified inaccuracy in Figure 6) and have highlighted the applicability of the developed mathematical models and simulation framework by supplementing article with the most relevant recently published applications:

  • Giulio, U.; Arents, J.Latella, A. AI in Industrial Machinery. Artificial Intelligence for Digitising Industry 2021.

  • Yuan, D.; Shu, X.; Liu, Q.; He, Z. Structural target-aware model for thermal infrared tracking. Neurocomputing 2022, 491, 44–56. 396, https://doi.org/https://doi.org/10.1016/j.neucom.2022.03.055.

  • Wei, Y.; Su, Z.; Mao, S.; Zhang, D. An Infrared Defect Sizing Method Based on Enhanced Phase Images. Sensors 2020, 20. 398, https://doi.org/10.3390/s20133626.

  • Sousa, M.J.; Moutinho, A.; Almeida, M. Thermal Infrared Sensing for Near Real-Time Data-Driven Fire Detection and Monitoring 400, Systems. Sensors 2020, 20. https://doi.org/10.3390/s20236803.

  • Zhuo, L.; Yang, X.; Zhu, J.; Huang, Z.; Chao, J.; Xie, W. Size determination of interior defects by reconstruction of subsurface 402 virtual heat flux for step heating thermography. NDT & E International 2023, 133, 102734. https://doi.org/https://doi.org/10.101 4036/j.ndteint.2022.102734.

  • Jensen, F.; Terlau, M.; Sorg, M.; Fischer, A. Active Thermography for the Detection of Sub-Surface Defects on a Curved and 405 Coated GFRP-Structure. Applied Sciences 2021, 11. https://doi.org/10.3390/app11209545.

Reviewer 2 Report

1. What are the applications of your work in real time 

2. In figure 7, each image description should be presented clearly

3. Table 1 and table 2, each image description should be clearly presented 

4. Refer latest research papers, regarding this work. 

5. How other noises can be minimized using this model, should be presented clearly 

Author Response

Thank you for your valuable recommendations and stimulating questions. Further, we clarify our responses.

Point 1: What are the applications of your work in real-time?

This work bolsters real-time performance 1) by supporting the development of new more efficient algorithms and 2) by leveraging more advanced acquisition and sensor degradation models to support the sensor’s online calibration using a range of input variables (sensor temperature, ambient temperature, degradation indicators, and others), i.e., aiding the development of shutter-less cameras. We have highlighted this aspect in the article accordingly.

Point 2: In figure 7, each image description should be presented clearly

Thank you, we update the figure to incorporate the corresponding blackbody temperatures into the images.

Point 3: In table 1 and Table 2, each image description should be clearly presented

Both tables were modified by changing the designations, adding nominal values, and providing more detailed explanations.

Point 4: Refer latest research papers, regarding this work.

We have not identified more recent work about modeling IR acquisition systems, nonetheless, we have highlighted the applicability of the developed mathematical models and simulation framework by supplementing our article with the most relevant recently published articles on potential applications:

  • Giulio, U.; Arents, J.; .; Latella, A. AI in Industrial Machinery. Artificial Intelligence for Digitising Industry 2021.

  • Yuan, D.; Shu, X.; Liu, Q.; He, Z. Structural target-aware model for thermal infrared tracking. Neurocomputing 2022, 491, 44–56. 396, https://doi.org/https://doi.org/10.1016/j.neucom.2022.03.055.

  • Wei, Y.; Su, Z.; Mao, S.; Zhang, D. An Infrared Defect Sizing Method Based on Enhanced Phase Images. Sensors 2020, 20. 398, https://doi.org/10.3390/s20133626.

  • Sousa, M.J.; Moutinho, A.; Almeida, M. Thermal Infrared Sensing for Near Real-Time Data-Driven Fire Detection and Monitoring 400, Systems. Sensors 2020, 20. https://doi.org/10.3390/s20236803.

  • Zhuo, L.; Yang, X.; Zhu, J.; Huang, Z.; Chao, J.; Xie, W. Size determination of interior defects by reconstruction of subsurface 402 virtual heat flux for step heating thermography. NDT & E International 2023, 133, 102734. https://doi.org/https://doi.org/10.101 4036/j.ndteint.2022.102734.

  • Jensen, F.; Terlau, M.; Sorg, M.; Fischer, A. Active Thermography for the Detection of Sub-Surface Defects on a Curved and 405 Coated GFRP-Structure. Applied Sciences 2021, 11. https://doi.org/10.3390/app11209545.

Point 5: How other noises can be minimized using this model, should be presented clearly

Thank you for pointing out this critical article’s claim. To our understanding other noise sources can be accounted for by improving 1) models to further advance physical sensor structure (e.g., cap design, heat dispersion, consequences of the vacuum quality) and 2) readout electronics for increased sensitivity and enhanced algorithm design (e.g., explore temporal noise filtering while increasing sensor's readout frequency and employing multi-frame analysis).